# PartnerMAS: An LLM Hierarchical Multi-Agent Framework for Business Partner Selection on High-Dimensional Features

## Abstract

High-dimensional decision-making tasks, such as business partner selection, involve evaluating large candidate pools with heterogeneous numerical, categorical, and textual features. While large language models (LLMs) offer strong in-context reasoning capabilities, single-agent or debate-style systems often struggle with scalability and consistency in such settings. We propose PartnerMAS, a hierarchical multi-agent framework that decomposes evaluation into three layers: a Planner Agent that designs strategies, Specialized Agents that perform role-specific assessments, and a Supervisor Agent that integrates their outputs. To support systematic evaluation, we also introduce a curated benchmark dataset of venture capital co-investments, featuring diverse firm attributes and ground-truth syndicates. Across 140 cases, PartnerMAS consistently outperforms single-agent and debate-based multi-agent baselines, achieving up to 10–15% higher match rates. Analysis of agent reasoning shows that planners are most responsive to domain-informed prompts, specialists produce complementary feature coverage, and supervisors play an important role in aggregation. Our findings demonstrate that structured collaboration among LLM agents can generate more robust outcomes than scaling individual models, highlighting PartnerMAS as a promising framework for high-dimensional decision-making in data-rich domains. Our implementation is available at this anonymous link.

## 1 Introduction

In real-world decision-making, practitioners often navigate high-dimensional data including extensive option sets and numerous evaluative features (Sandanayake et al., 2018; Sigle et al., 2023). Business partner selection which includes partner shortlisting and strategic alliance formation exemplifies this challenge (Mindruta et al., 2016): firms often face a vast pool of potential candidates, each described by diverse attributes ranging from quantitative indicators (*e.g.*, financial metrics, geographic presence) to text-rich information (*e.g.*, strategic fit, investment preferences) (Shah & Swaminathan, 2008). The scale and complexity of such data can easily overwhelm human decision-makers, incurring significant costs (Li et al., 2008). This underscores the need for intelligent systems capable of analyzing large candidate sets and diverse features.

Large language models (LLMs) have emerged as promising tools for addressing reasoning tasks in data-rich domains (Lee et al., 2025; Mischler et al., 2024). Unlike traditional machine learning algorithms that often demand extensive training data, LLMs draw on pretrained knowledge and in-context reasoning to interpret heterogeneous information (Li et al., 2025a). With appropriate prompting (*e.g.*, few-shot learning) or information retrieval techniques (*e.g.*, RAG), these models can identify salient features using only feature and task descriptions, achieving performance comparable to established methods (Li et al., 2025a; Jeong et al., 2024). As task complexity has increased across domains, researchers have increasingly moved beyond single-agent approaches toward multi-agent systems (MAS), wherein complex problems are decomposed into specialized sub-tasks managed by agents operating within structured collaborative workflows (Li et al., 2024). This enables more sophisticated problem-solving by distributing cognitive load across multiple specialized components, each optimized for specific aspects of the overall task. Recent studies have demonstrated

the effectiveness of MAS across diverse domains, including software development (Tao et al., 2024), mathematical reasoning (Li et al., 2025b), and healthcare decision-making (Chen et al., 2025).

Despite these advances, a significant gap remains in the application of MAS to high-dimensional decision-making tasks within high-stakes domains such as finance, where effective automation could substantially reduce cognitive burden on human experts while improving decision quality. Current research on financial MAS has concentrated primarily on a narrow range of applications, such as individual stock trading (Yu et al., 2024) and portfolio management (Luo et al., 2025), leaving numerous critical financial decision-making areas substantially underexplored. This study addresses business partner selection as a representative example of such underexplored domains, where MASs need to deal with high-dimensional, heterogeneous features. This setting requires both scalability and nuanced feature reasoning, a new aspect that previous MAS works have not explored much.

To investigate these challenges and advance MAS design, we develop a hierarchical MAS framework, **PARTNERMAS** for business partner selection on high-dimensional features. PARTNERMAS follows a three-tier design: a *Planner Agent* first analyzes the investment context and creates specialized evaluators; multiple *Specialized Agents* then assess candidate firms from different perspectives; finally, a *Supervisor Agent* integrates their outputs to make the final selection. This hierarchical design brings several advantages: it enables decomposition of complex decision-making tasks, allows weaker agents to contribute effectively through specialization, and provides robustness by synthesizing diverse perspectives rather than relying on a single model's judgment. To summarize, this research makes the following two key contributions: (i) We introduce a tabular benchmark for co-investor selection that captures real-world decision-making scenarios, featuring diverse candidate firms and multifaceted evaluation criteria. (ii) We design and implement PARTNERMAS that mirrors expert roles in business partner selection. Through extensive empirical validation, we demonstrate that PARTNERMAS achieves significant performance improvements of approximately 15% compared to single agent and traditional multi-agent debate methods.

## 2 RELATED WORK

High-dimensional data refers to datasets characterized by large numbers of heterogeneous features (Tang et al., 2016). These settings pose well-documented challenges (Johnstone & Titterington, 2009) for both traditional machine learning and LLM-based methods, including overfitting (Kim & Simon, 2014), feature redundancy (Ferreira & Figueiredo, 2012), and difficulties in interpreting model outputs (Potts & Schmischke, 2021). Such data is prevalent in high-stakes domains like healthcare (Patra et al., 2021) and finance (Fallahgoul, 2025).

**Business Partner Selection.** Selecting partners is a crucial first step in establishing business relationships (Shah & Swaminathan, 2008; Cummings & Demirkan). Prior research highlights central drivers of partner selection, including value creation and trustworthiness. Firms often evaluate potential partners based on complementary resources and capabilities like knowledge, technology, or capital, to assess whether collaboration generates greater value than independent efforts (Furlotti & Soda, 2018; Mindruta et al., 2016). Meanwhile, trustworthiness is essential to mitigate risks from opportunistic behavior (Das & Rahman, 2010; Li et al., 2008). Trust is often inferred from past collaborations or, in their absence, signals like reputation and transaction records (Lumineau et al., 2021). Beyond value and trust, firms also weigh coordination costs, which can arise from communication demands, geographical distance, or cultural differences, causing misunderstandings even with trustworthy partners (Gulati et al., 2012). To reduce risks, firms often prefer partners with geographic proximity, similar industry backgrounds, or cultural homophily (Mayer & Argyres, 2004). These considerations make partner selection a complex, multidimensional decision-making process.

In the venture capital (VC) sector, which is the focus of our study, co-investor selection often begins with a broad search for potential partners. After compiling an initial list, experienced managers carefully evaluate multiple factors to identify firms that align best with their objectives. The lead VC then extends invitations to the shortlisted candidates. However, this process is often lengthy and labor-intensive, and the costs of partner selection are recognized as a major ex-ante transaction cost in business exchanges (Lumineau et al., 2021), particularly for VC firms that frequently engage in collaborations. This challenge presents the need for more intelligent and efficient selection methods.

**LLM-driven Feature Selection.** By leveraging pretrained knowledge, LLMs can rank, filter, or explain the importance of features using their names and task context (Jeong et al., 2024). Jeong et al. (2024) has developed pipelines where LLMs like GPT-4 can generate feature importance scores or

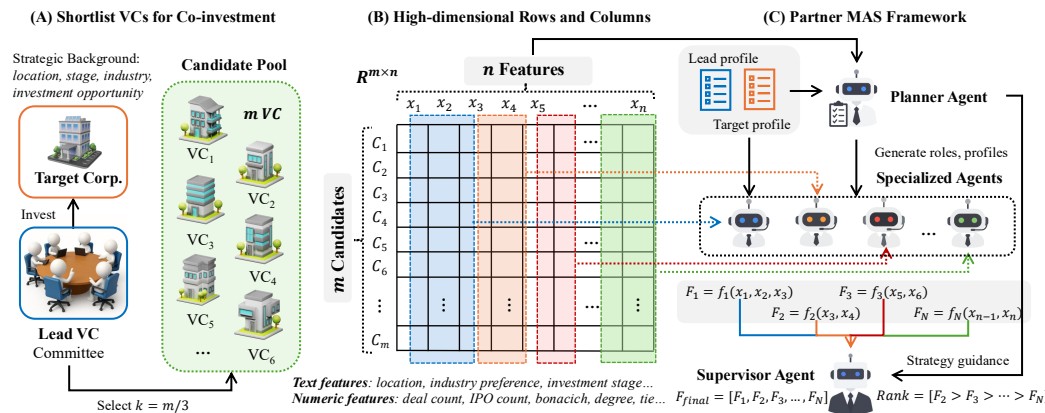

Figure 1: An illustration of the research design. (A) Co-investors shortlisting. (B) High-dimensional feature selection. (C) Hierarchical MAS framework.

explanations. Li et al. (2025a) has shown that LLMs can reliably identify key predictors in domains like healthcare and finance. In biomedical settings, researchers have improved performance by retrieving definitions of gene or protein identifiers to ground LLM reasoning in domain knowledge (Lee et al., 2025). Such LLM-driven selectors can rival standard statistical feature selection techniques (*e.g.* LASSO) even in zero-shot settings (Zhang et al., 2025).

Beyond single LLM, researchers have increasingly turned to MAS to handle complex decision-making. In MAS, agents assume specialized roles, such as planner, critic, or domain expert, and interact through structured workflows. For instance, MetaGPT assigns LLM agents to emulate software development teams (Hong et al., 2024), while debate-style frameworks allocate agents to critique and refine reasoning in math and coding tasks (Chan et al., 2024; Liang et al., 2024). Similar approaches in healthcare show that a "generalist" can triage cases and delegate to specialists for targeted diagnosis (Zuo et al., 2025). Such architectures outperform single LLMs by combining division of labor with consensus and conflict resolution. Some recent work applies MAS to feature engineering, where selector, generator, and coordinator agents refine feature sets (Gong et al., 2025). However, applying MAS to high-dimensional business data still remains underexplored, and little evidence shows how role-specialized agents can reliably coordinate to produce feature-informed decisions. This gap motivates our exploration of LLM-based MAS for business partner selection.

## 3 METHODOLOGY

### 3.1 PROBLEM FORMULATION

Figure 1 illustrates our MAS architecture design. We specifically examine how the lead VC evaluates the initial candidate pool to create a more targeted shortlist (Figure 1 (A)). We do not address the final stage of selection, which moves from the shortlist to the actual partnership, as it involves more nuanced negotiation processes between the parties. By focusing on the shortlisting phase, we investigate the potential of LLM-based MAS to support the partner selection process.

We consider the candidate pool as high-dimensional, heterogeneous tabular data with $m$ candidates (rows) and $n$ features (columns) (Figure 1 (B)). Let it be $C = \{c_1, \ldots, c_m\}$, where each candidate $c_j$ is represented by a feature vector $x_j \in \mathbb{R}^n$. The feature matrix is $X \in \mathbb{R}^{m \times n}$. Columns are mixed-type consisting of both text fields (*e.g.*, investment preference, firm type, industry preference) and numeric attributes (*e.g.*, deal count, IPO count, network degree, tie strength). Given a task context $\mathcal{Q}$ (*e.g.*, lead VC and target company profiles), the goal is to produce a shortlist of candidates:

$$S = (s_1, \ldots, s_k), \quad S \in C^k, \quad k = \lfloor m/3 \rfloor, \tag{1}$$

*i.e.*, the top $\lfloor m/3 \rfloor$ candidates from the pool. The proportion is set to one-third to emulate a realistic VC screening process, where a large pool of candidates is progressively narrowed down. For instance, an initial pool of 36 candidates is filtered to a shortlist of 12, from which a final group of four co-investors is selected (feature description detailed in Appendix C).

### 3.2 MAS ARCHITECTURE DESIGN

To tackle the challenge of partner selection, we introduce a hierarchical MAS framework called **PARTNERMAS** (Figure 1 (C)). This design enables the system to decompose the high-dimensional

task into manageable sub-tasks, each assessing tabular data from different dimensions: (i) **Planner Agent** ($PA$) that designs the evaluation strategy. (ii) **Specialized Agents** ($SA$), $\{SA_1, \ldots, SA_N\}$, which are dynamically configured by the $PA$ to execute this strategy based on their role definition and expertise. (iii) **Supervisor Agent** ($SPA$) that aggregates analyses to make a final decision.

**Planner Agent** ($PA$). The primary role of $PA$ is to interpret the high-level task context $\mathcal{Q}$ (Lead VC and target company profiles) and formulate an evaluation plan. It does so by analyzing $\mathcal{Q}$ with feature names to identify the most critical evaluation dimensions given the case description. The $PA$'s output is a set of $N$ agent configurations, $\{A_1, \ldots, A_N\}$, where each configuration $A_i$ contains a specific "profile" that guides the corresponding $SA_i$. It also generates "strategic guidance" to support the decision for the $SPA$. Formally, the output can be expressed as: $[\{A_1, \ldots, A_k\}, PA(\mathcal{Q}, C_{sample})]$, a set of explicit instructions for the team of $SA$s.

**Specialized Agent** ($SA$). Each $SA$ acts as a domain expert, tasked with evaluating the entire candidate pool $C$ from its specialized perspective. To manage the high dimensionality of the candidate feature vectors ($x_j \in \mathbb{R}^n$), the $SA$ first performs a feature selection step. Guided by its assigned $profile_i$, it identifies and focuses on a relevant feature subset. The agent's core evaluation function $f_i$, driven by a backbone LLM, then directly produces a ranked shortlist of candidates. This output, $S'_i$, is a list containing the top $k' = \lceil m/3 \rceil$ firms, where each entry includes the firm's ID, its rank, and an alignment score, $score_{ij} \in [1, 10]$.

**Supervisor Agent** ($SPA$). The $SPA$ is responsible for synthesizing the shortlists $\{F'_1, \ldots, F'_N\}$ from Specialized Agents into the final ranked list $F$. It mimics a human-led committee's decision-making process by first establishing consensus and then resolving disagreements based on the strategic priorities of the deal. This is achieved in a two-step process: (i) **Consensus Selection**: The $SPA$ first identifies candidates with broad support across $SA$, by counting how many agents include them in their shortlists. This step can determine robust candidates that perform well across different dimensions: $F_1(c_j) = \sum_{i=1}^{N} \mathbb{I}(c_j \in F'_i)$. (ii) **Conflict Resolution**: To fill the remaining slots in the shortlist, the $SPA$ resolves disagreements among the $SA$s. It first determines an importance weight $w_i$ for each agent based on the agent's relevance to the specific deal. It then resolves the conflict for the remaining candidates by giving more weight to the opinions of more important $SA$s. This allows the $SPA$ to select candidates who excel in critical areas, even if they lack broad consensus: $F_2(c_j) = \sum_{i=1}^{N} w_i \cdot \frac{1}{R_i(c_j)}$. The final shortlist $F = [F_1(c_j), F_2(c_j)]$.

### 3.3 Evaluation

The system's performance is evaluated against a ground truth of successful co-investor partnerships. For each task, the generated shortlist $F$ is compared to the set of ground-truth co-investors, $G$. The primary metric is the **Match Rate**, which measures the fraction of actual partners that are successfully identified by the MAS. It is formally defined as the recall of the system:

$$\text{Match Rate} = |F \cap G|/|G| \times 100\%. \tag{2}$$

For instance, if the initial candidate pool has 36 VC firms ($m = 36$), our PartnerMAS generates a shortlist of 12 ($k = 12$). If the ground truth contains 4 actual co-investors ($|G| = 4$) and 3 are found within the system's shortlist of 12 ($|F \cap G| = 4$), the Match Rate is 75%. This metric measures the system effectiveness by ensuring that the true positive candidates are included in the final shortlist.

## 4 Experiment Design

### 4.1 Data Preparation

Our primary data source is the London Stock Exchange Group (LSEG) Workspace (London Stock Exchange Group, 2024), from which we collect VC investment records from 1980 to 2024. Following prior VC research (Makarevich, 2018; Wang et al., 2022), we restrict the sample to U.S.-based companies to avoid confounding effects from cross-country differences in regulatory frameworks and market institutions. We further exclude solo investments to focus on co-investments, where collaboration among investors is observable. After this initial filtering, the dataset comprise 52,662 companies backed by 16,030 VC firms, and we identify all active VCs in each year, industry, and state to construct the relevant candidate pool for each year–state–industry context.

To examine multiparty syndicate formation, specifically the process by which lead VCs select co-investment partners, we further restrict the sample to companies with complete first-round information and syndicates of at least three investors. We then merge the company list from LSEG with

PitchBook (PitchBook Data, Inc., 2024) to identify the lead VC for first-round investment based on the company name and the headquarter state. Only matches labeled as "high" or "very high" confidence are retained to ensure accurate lead investor identification. For analytical tractability, we limit the sample to cases with a single lead VC, since multiple leads often involve more complex governance and negotiation dynamics (Lerner, 2022; Kaplan & Strömberg, 2003). After these filters, the sample include 2,218 companies. We then merge PitchBook data back into the LSEG sample to retrieve lead VC information, supplemented by manual matching (company name, VC firm name, industry, and state) and excluding samples without necessary information. Applying all restrictions yields a final dataset of 140 cases for subsequent [1].

## 4.2 Experiment Settings

We evaluate multiple experiments for partner selection. All experiments use the same dataset and evaluation protocol, and all LLMs are run with temperature set to 0 to minimize output variance.

**Baseline Configurations: Single Agent.** In this baseline, a single LLM agent reviews all candidate firms and produces a ranked shortlist without external feedback. The parameter $k$ denotes the number of independent runs: $k=1$ corresponds to a one-shot evaluation, while $k > 1$ allows the agent to generate multiple candidate shortlists. To obtain a final decision, the agent engages in a self-reflection step, comparing its own $k$ outputs and selecting the one it deems most reliable. This design tests both the limitations of single-pass reasoning and the potential benefits of repeated deliberation within a single-agent framework. Unless otherwise specified, we set $k=1$ by default. The details for the Single Agent prompts are shown in Appendix F.2.

**Baseline Configurations: Debate MAS.** The second baseline implements a debate-based multi-agent system (MAS) inspired by prior work (Chan et al., 2024; Liang et al., 2024). Three specialized agents simulate a VC committee: each independently evaluates candidates, critiques peers' reasoning while scores remain hidden, and then revises its judgments in light of feedback. A supervisor agent synthesizes their inputs into a final shortlist. The details for the Debate MAS design and prompts are shown in Appendix E and Appendix F.3.

**Agent Configuration: PARTNERMAS .** Our MAS adopts the Planner–Specialist–Supervisor design described in Section 3. Unlike Debate MAS, which emphasizes adversarial critique, PARTNERMAS is built on structured collaboration and coordinated division of labor. Its design varies along two main dimensions: prompt guidance and backbone assignment. For prompts that guide Planner Agent and Supervisor Agent, we compare two conditions: (i) generic prompts without business knowledge, and (ii) business-domain guided prompts, which encourage agents to explicitly consider dimensions, including collaboration networks, industry fit, financial capacity, and geography (details in Appendix F). The details for the PARTNERMAS prompts are shown in Appendix F.4.

## 5 Experimental Results

### 5.1 Performance Benchmarking

We first compare the overall performance of PARTNERMAS against Single Agent and debate-based baselines with regard to the overall match rate across all business cases in our dataset. Results are summarized in Figure 2. Our key observations and analysis are as follows.

**PARTNERMAS achieves the strongest results.** Our hierarchical multi-agent system consistently outperforms all baselines. For example, PARTNERMAS with gpt-4.1-mini as the backbone achieves a match rate of 70.89% with business-domain guided prompt, which exceeds the best-performing single LLMs, such as gpt-5 (medium effort) (61.50%) and gemini-2.5-pro (61.42%). Notably, gpt-4.1-mini is a smaller and more cost-efficient model, its token cost is roughly an order of magnitude lower than gpt-5 or gemini-2.5-pro, yet when embedded in PARTNERMAS it delivers markedly stronger outcomes. This pattern holds even when using smaller backbones like gpt-5-nano, where PARTNERMAS still delivers 8–10% higher match rates than the same model in a Single Agent configuration. These results underscore that coordination among specialized agents can compensate for and often surpass pure scaling of model size. Importantly, the gains remain robust across different backbone LLMs, confirming that our framework is not tied to single LLMs.

---

[1]Due to intellectual property constraints of VC firms, the dataset cannot be released publicly but is available from the authors upon request for pure research purposes.

**Debate alone does not guarantee improvements.** We next assess whether adversarial interaction between agents helps. The debate-based MAS baseline sometimes produces modest gains relative to Single Agent baselines (*e.g.*, 57.12% for gpt-5-nano in Debate MAS vs. 55.14% for its Single Agent counterpart). However, the overall effect is inconsistent: for gpt-4.1-mini, Debate MAS reaches only 60.19%, which lags substantially behind PARTNERMAS with the same backbone, and in some cases performance even drops below the Single Agent level. A likely explanation is that debate can distract agents from their original reasoning or amplify minor errors, rather than directing to stronger solutions. This finding shows that while debate can reveal reasoning errors, it lacks the structured role division and aggregation mechanisms needed for complex business tasks.

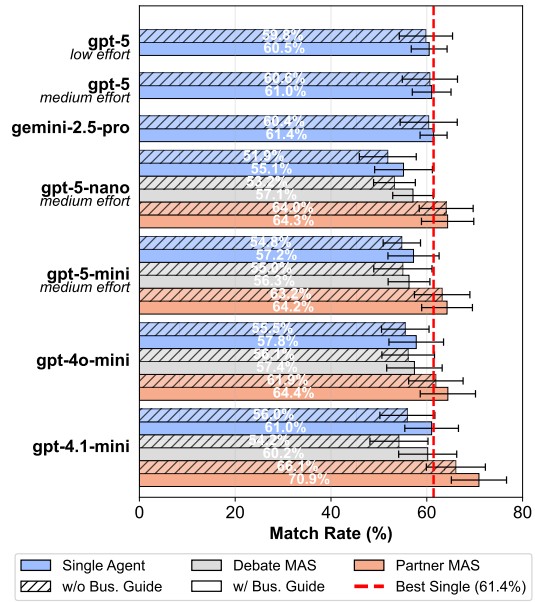

Figure 2: Performance benchmark for Single Agent, Debate MAS, and PARTNERMAS .

**Business-domain guidance helps.** To verify the effect of business domain knowledge, we conduct experiments comparing model performance with and without business-domain guidance. The text for the guidance is shown in Appendix F.1. Across nearly all model families, introducing business domain guidance leads to consistent accuracy gains, although marginal. For Single Agent baselines, business-domain guidance improves match rates by 2–5% absolute, such as for gpt-4.1-mini (55.95%). The improvements are more substantial for PARTNERMAS , with some configurations showing gains of over 7%, rising from 62.55% to 69.03%). These results demonstrate that grounding agents in domain-relevant dimensions—such as financial capacity, collaboration networks, or geographic compatibility—substantially enhances shortlist quality. Interestingly, the magnitude of improvement varies across models, indicating that stronger backbones are better able to exploit domain cues, while smaller models benefit but plateau earlier.

**Backbone LLM effects.** Table 5.1 shows that among Single Agent baselines, gpt-4.1-mini achieves the highest match rates when business domain guidance is provided, followed by gpt-4o-mini, while gpt-5-nano (medium effort) generally lags behind. This suggests medium-sized models balance reasoning ability and efficiency, whereas smaller models underperform and larger ones may not justify their cost. When incorporated into PARTNER-

Table 1: Match rates (mean ± 95% CI) across different settings. All gpt-5-nano adopt the medium thinking effort.

| PA | SA | SPA | Match Rate |
|---|---|---|---|
| gpt-4o-mini | gpt-4o-mini | gpt-4o-mini | 64.40% ± 5.77 |
| gpt-4.1-mini | gpt-4o-mini | gpt-4o-mini | 63.21% ± 5.81 |
| gpt-5-nano | gpt-4o-mini | gpt-4o-mini | 65.19% ± 5.89 |
| gpt-4o-mini | gpt-4.1-mini | gpt-4o-mini | 62.14% ± 6.03 |
| gpt-4o-mini | gpt-5-nano | gpt-4o-mini | 64.70% ± 5.48 |
| gpt-4o-mini | gpt-4o-mini | gpt-4.1-mini | 69.03% ± 5.94 |
| gpt-4o-mini | gpt-4o-mini | gpt-5-nano | 64.70% ± 5.48 |

MAS , however, even lightweight models like gpt-5-nano (medium effort) gain from structured role division, achieving 8–10% higher match rates relative to their Single Agent counterparts. These findings highlight that the hierarchical design not only amplifies the capabilities of stronger backbones but also compensates for the weaknesses of smaller ones, yielding robust improvements.

**Performance–efficiency tradeoff.** Figure 3 illustrates the trade-off between match rate and the overall token consumption. To ensure fairness, we evaluate Single Agent baselines with $k{=}4$ runs, aligning their computational cost with the multiple agents in Debate MAS (3+1 agents) and PARTNERMAS (average of 4.27 agents). The results show that large single models such as gpt-5 and gemini-2.5-pro fall into a high-cost yet only moderate-accuracy regime. In contrast, PARTNERMAS configurations consistently deliver both higher match rates and lower token budgets. For example, PARTNERMAS with gpt-4.1-mini achieves over 70% accuracy while consuming fewer tokens than

gpt-5 (medium effort). This efficiency arises from decomposition: specialized agents narrow their focus to smaller feature subsets, reducing redundancy and improving coordination. Consequently, PARTNERMAS proves not only more accurate but also more cost-effective, making it well suited for practical deployment where API costs and inference latency are critical constraints.

## 5.2 AGENT PERFORMANCE AND REASONING ANALYSIS

To understand the internal dynamics, we first analyze agent performance (Figure 4) and then use regression models to examine reasoning at each layer of the PARTNER-MAS hierarchy. We evaluate the Planner Agent's deployment strategy with logistic and linear regression (Table 2), visualize Specialized Agents' performance and feature focus with heatmaps (Figure 5), and assess the Supervisor Agent's decision-making through regression analysis (Table 3). This analysis reveals how different backbones and business-domain guidance, shape agent reasoning and overall performance.

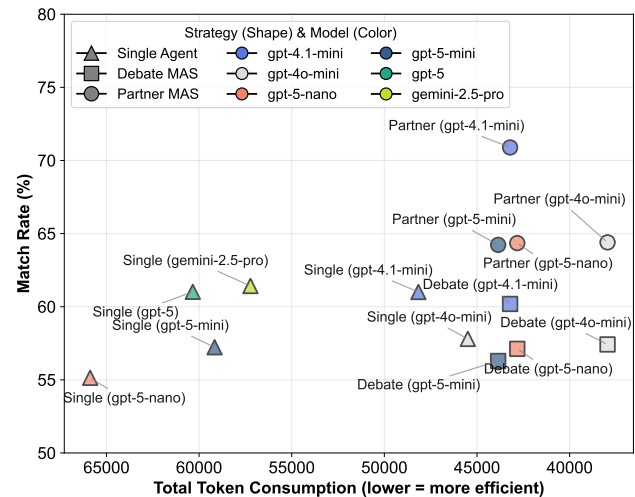

Figure 3: Model performance comparison across settings.

**System and agents performace**. Figure 4 shows the performance of Specialized Agent clusters (see Appendix D for details on how we handle variations in Specialized Agent names) on top right corner (achieving 65% to 75% accuracy with less than $45k$ token usage), indicating our multi-agent-system build not only achieve higher accuracy but is also more effective. To better understand how different agents perform and contribute to the final success of the multi-agent-system, we present Figure 4: sub-figure (A) relates cluster performance to their usage and importance. Partnership History and Network Connectivity agents tend to be among the strongest performers overall while Industry and Geographic agents are often ranked highly by the planner/supervisor despite delivering comparatively lower accuracy. This misalignment implies that the planning and supervision logic may be overvaluing broad topical coverage relative to historically effective signals for co-investment discovery. Sub-figure (B) and (C) show that MAS accuracy is highest when the number of active agents is modest (approximately 4–5) and opinion diversity is more concentrated (lower normalized HHI). As agent count and heterogeneity grow, returns diminish and aggregation becomes harder, consistent with the hypothesis that excessive diversity can dilute the supervisor's ability to extract a coherent signal.

**Model choice and business-domain-guided prompt drive planning.** Our analysis in Table 2 reveals that the Planner's decisions are most significantly influenced by its core instructions and backbone LLM rather than specific case context. The backbone LLM and the inclusion of a business domain guidance are overwhelmingly the strongest predictors for the type and number of agents deployed. For example, the odds of deploying an "Industry & Sector" agent increase by a factor of 57.61 when a business domain guidance is provided. In contrast, for most cases, the contextual factors, such as the target company's industry focus or geolocation, show no statistically significant effect. This indicates that the Planner Agent operates at a strategic level, relying on its guiding prompts and model architecture to structure the problem-solving approach, rather than reacting to the fine-grained details of each case.

**Specialized Agents value different features and their performance varies.** As illustrated in Figure 5, the effectiveness of Specialized Agents highly depend on their assigned role, leading to significant performance variations across different areas of expertise. For instance, when using the gpt-4.1-mini backbone, the "Risk & Compliance" agent excels with an impressive 83.3% accuracy, while the Investment Stage" agent struggles, achieving only 37.7%. Similar gaps appear with other models, though those driven by gpt-5-nano exhibit noticeably lower variance.

This performance difference is probably explained by how agents, guided by their roles, focus on different features. Agents can successfully identify and prioritize relevant information; for exam-

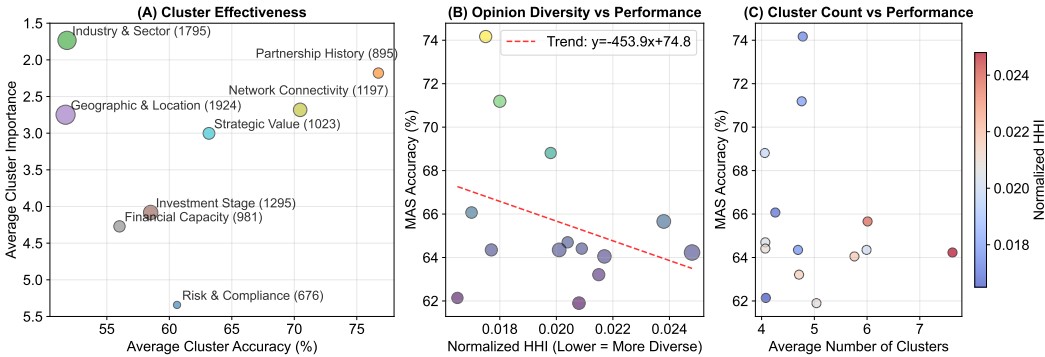

Figure 4: Agent performance grouped by Specialized Agent clusters. (A) Bubble chart of eight agent clusters, in which accuracy (x) vs importance rank (y, 1=highest), Bubble size = agents per cluster. (B) PARTNERMAS accuracy vs normalized HHI (lower = more diverse) with trend line. Node size = cluster count. (C) PARTNERMAS accuracy vs number of Specialized Agent clusters.

Table 2: Planner Agent deployment regression analysis with label encoder.

| Variable | Agent Presence (Logistic Regression - Odds Ratios) | | | | | | | | Agent Count |
|---|---|---|---|---|---|---|---|---|---|
| | Risk & Compliance | | Industry & Sector | | Financial Capacity | | Strategic Value | | Linear Regression |
| | OR | Sig. | OR | Sig. | OR | Sig. | OR | Sig. | Coef. (SE) |
| prompt_hint | 23.06 | *** | 57.61 | *** | 2.49 | *** | 1.16 | n.s. | 0.12** (0.05) |
| model | 141.94 | *** | 0.55 | *** | 2.47 | *** | 6.52 | *** | 0.95*** (0.02) |
| company_state | 1.02 | n.s. | 1.00 | n.s. | 1.02 | n.s. | 0.99 | n.s. | -0.00 (0.00) |
| company_industry | 0.71 | * | 1.01 | n.s. | 0.91 | n.s. | 1.09 | n.s. | -0.03 (0.03) |
| firm_investment_stage | 1.02 | n.s. | 1.00 | n.s. | 0.96 | * | 1.00 | n.s. | -0.00 (0.01) |
| firm_type | 0.88 | n.s. | 1.16 | * | 1.05 | n.s. | 0.88 | n.s. | -0.01 (0.02) |
| geography_preference | 0.98 | n.s. | 1.00 | n.s. | 0.97 | *** | 1.03 | ** | -0.00 (0.00) |
| firm_state | 0.99 | n.s. | 1.01 | n.s. | 0.98 | n.s. | 0.99 | n.s. | 0.00 (0.00) |
| firm_industry_focus | 0.99 | n.s. | 1.01 | n.s. | 1.01 | n.s. | 1.02 | n.s. | -0.00 (0.00) |
| $R^2$ | 0.711 | | 0.302 | | 0.181 | | 0.361 | | 0.664 |

**Notes:** prompt_hint (generic = 0; business = 1). model (gpt-4o-mini = 0; gpt-4.1-mini = 1; gpt-5-nano = 2; gpt-5-mini = 3). Significance: *** $p < 0.001$, ** $p < 0.01$, * $p < 0.05$, n.s. $p \geq 0.05$. Other four Specialized Agent clusters do not show any significance and therefore are not presented in this table.

ple, the Geographic & Location" agent correctly emphasizes "geography_preference" and "location" features. Interestingly, the underlying LLM model shapes the agent's reasoning style. The "gpt-4.1-mini" model demonstrates a sharp, direct focus on specific features, with the Industry & Sector" agent targeting firm industry focus and the "Network Connectivity" agent concentrating on tie strength, degree, and bonacich centrality. In contrast, the gpt-5-mini model displays a more distributed pattern. This is because a Planner Agent driven by gpt-5-mini or -nano tends to generate a larger number of Specialized Agents, whose abilities may overlap. However, this broader agent deployment does not translate to superior performance. The performance of gpt-5-nano-driven agents is distributed, with no cluster exceeding 70% accuracy. This trend is also observable for gpt-5-mini, despite a standout 92.5% accuracy from its "Partnership History" agent cluster.

**Prioritize the right expert can increase the performance.** The regression analysis in Table 3 investigates how the Supervisor's ranking of agent importance correlates with the final match rate. For the gpt-5-nano backbone, assigning the top rank to the "Partnership History," "Industry & Sector," or "Geographic & Location" agent is a highly significant predictor of a correct final outcome (p¡0.001). For gpt-4.1-mini, success is strongly correlated with prioritizing agents on "Investment Stage" ($p < 0.001$), "Strategic Value" ($p < 0.01$), and "Network Connectivity" ($p < 0.01$). This highlights that the Supervisor's ability to weigh expert opinions is a key determinant of the system's performance. Even if individual Specialized Agents perform well, an error in prioritization by the Supervisor can lead to a suboptimal final shortlist.

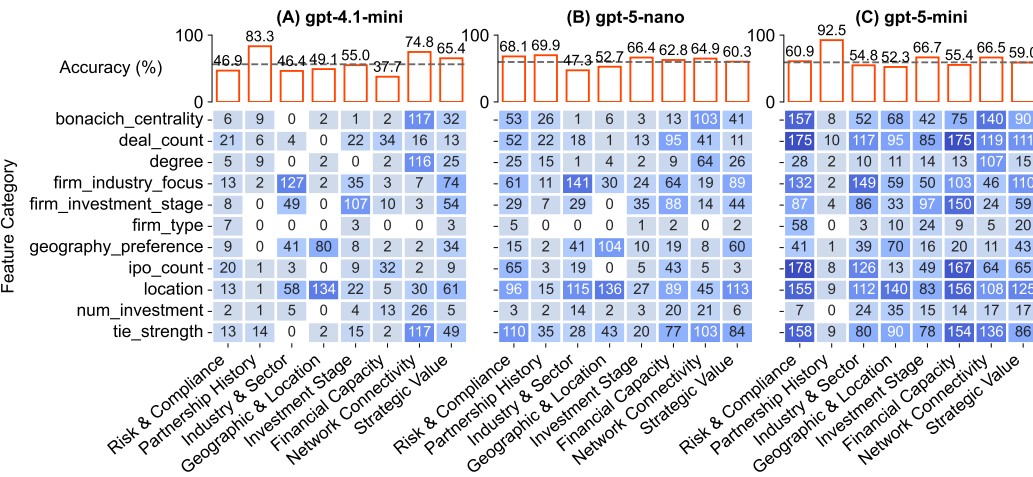

Figure 5: Accuracy and feature focus of Specialized Agents under different backbones: (A) gpt-4.1-mini. (B) gpt-5-nano. (C) gpt-5-mini.

## 6 CONCLUSION AND DISCUSSION

Our work makes three key contributions. First, we introduce a benchmark dataset for evaluating LLMs on high-dimensional tabular problems that combine numerical, textual, and categorical attributes. Grounded in real VC co-investment records, the dataset provides a realistic and challenging setting for testing reasoning in structured business decisions. Second, we propose PARTNERMAS ,

Table 3: Regression analysis of Supervisor Agent's ranking.

| Variable | gpt-4.1-mini | gpt-4o-mini | gpt-5-nano |
|---|---|---|---|
| R1_Risk & Compliance | - | - | -40.38 |
| R1_Partnership History | 7.29 | 21.15 | 20.33*** |
| R1_Industry & Sector | 2.39 | 16.04 | 23.44*** |
| R1_Geographic & Location | 20.21* | 32.78 | 20.10*** |
| R1_Investment Stage | 22.43*** | - | 13.59 |
| R1_Network Connectivity | 11.98** | - | 20.69*** |
| R1_Strategic Value | 19.35** | -15.21 | 13.59 |
| prompt_hint_business | 46.77*** | 28.50 | 33.71*** |
| prompt_hint_generic | 36.89*** | 25.00 | 37.64*** |
| $R^2$ | 0.040 | 0.034 | 0.057 |

**Notes.** prompt_hint_business (yes =1; no =0). prompt_hint_generic (yes = 1; no = 0). Significance: *** $p < 0.001$, ** $p < 0.01$, * $p < 0.05$.

a hierarchical MAS that decomposes complex evaluation into planner, specialist, and supervisor layers. Unlike prior single-agent or debate MAS approaches, PARTNERMAS can dynamically configure role-specialized agents and coordinate their outputs to improve shortlist accuracy. While our case study focuses on business partner selection, the framework's reliance on in-context reasoning rather than task-specific training suggests potential applicability to other domains, though broader validation is still required. Third, our analyses reveal how different layers contribute to overall performance: planners are most responsive to business-domain guidance, specialized agents generate complementary perspectives that improve coverage, and supervisors play a decisive role in integrating these signals into consistent final outcomes.

This study also presents several opportunities for future work. The dataset remains comparatively small due to availability and quality constraints, and its focus on U.S. venture capital restricts the test size. Our evaluation also relies primarily on advanced GPT backbones, leaving open how the system performs with lighter or open-source models that would be more practical in resource-constrained environments. Finally, while specialized agents often achieve strong role-specific accuracy, the Supervisor Agent occasionally aggregates them poorly, reducing final shortlist quality. This suggests that coordination, not specialization, is the current bottleneck; improving supervisory mechanisms through meta-reasoning or structured consensus remains a key direction for future research. Overall, PARTNERMAS demonstrates that structured collaboration among LLM agents can outperform both single-agent and debate-based MAS baselines for high-dimensional decision-making. These results highlight the promise of PARTNERMAS as a general framework for complex decision tasks, with implications for business, healthcare, and other data-rich domains.

## 7 ETHICS STATEMENT

We adhere to the ICLR Code of Ethics and conducted this study using company- and firm-level investment records from licensed sources (LSEG Workspace and PitchBook). The research does not involve human subjects or personally identifiable information. Due to licensing and confidentiality constraints associated with these data sources, we do not redistribute the raw data; we report aggregated analyses in the paper, and the curated test dataset is available from the authors upon reasonable request for research purposes only.

PARTNERMAS is designed as a decision-support system that produces shortlists and rationales to assist expert judgment; it is not intended to autonomously make or execute investment decisions. As with any system trained or powered by large language models and historical records, outputs may reflect biases or limitations present in data and models. We do not claim to have performed formal fairness or bias audits in this work; instead, we acknowledge this as an important limitation and encourage careful human oversight in any practical use. We reduce variance where possible by using deterministic settings (*e.g.*, temperature set to 0; see Section 4.2) and by grounding prompts in domain-relevant factors (Appendix F). We also describe our use of LLMs for research assistance (*e.g.*, coding help, grammar checks) in the Appendix (Use of LLMs).

## 8 REPRODUCIBILITY STATEMENT

We aim to make our study reproducible within the constraints of data licensing and LLM services. The problem setup, model architecture, and evaluation protocol are described in Section 3. Data construction and filtering steps for the candidate pool are detailed in Section 4.1, and the feature set is summarized in Appendix C. Experimental settings (including shared protocols and deterministic decoding with temperature set to 0) are provided in Section 4.2. The prompts used for each agent and configuration are included in Appendix F, including concrete versions for the Single Agent (Appendix F.2) and PARTNERMAS (Appendix F.4).

The code base for the project is available at `https://anonymous.4open.science/r/Partner-MAS-7DCE`. Our curated test dataset contains firm- and deal-level information subject to license and confidentiality obligations; to protect the privacy and commercial sensitivities of the companies involved, the dataset is available from the authors upon request for research purposes. The prompts necessary to reproduce the agent behaviors are appended at the end of the paper.

Due to the evolving nature of hosted LLM services and provider-side updates, exact numerical results may exhibit minor variation across runs or over time, even with temperature set to 0. To mitigate this, we standardize the evaluation metric (Match Rate), use a fixed protocol across all experiments, and document all key choices in the paper and appendix so that researchers can follow the same setup and compare results under similar conditions.

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

## A    USE OF LLMs

During the development of this paper, we use transformer based large language models in the following aspects:

- Reference discovery: use the deep research tools from major providers to explore relevant work and literature.
- Code assistance: use coding agents to assist developing the code base of the current work.
- Grammar check: use LLMs to detect grammar errors in the drafty version of the paper, for better displaying our results.

## B    DATA AVAILABILITY

This curated dataset offers a reliable foundation, as both LSEG (London Stock Exchange Group, 2024) and PitchBook (PitchBook Data, Inc., 2024) provide clear, consistent records of first-round investments and designated lead VCs, which allows researchers to cross-verify records.. While licensing restrictions prevent us from sharing the raw data publicly, it can be accessed for research purposes upon reasonable request to the corresponding authors.

## C    FEATURE DESCRIPTION

For each company's lead investor, we extract the relevant subset of potential coinvestors from the VC pool based on the company's industry, headquarter state, and the first-round investment year. We then construct pairwise lead VC–VC firm observations to capture prior co-investment experiences (Bellavitis et al., 2020) and geographic distance between partners (Sorenson & Stuart, 2001; Gulati et al., 2012). In addition to these network measures, we obtain detailed firm-level characteristics from LSEG Workspace (London Stock Exchange Group, 2024), including firm location (county and state) and investment preference variables such as geographic, industry, and stage preferences. These additional attributes allow us to account for both structural and preference-based drivers of syndicate partner selection, as illustrated in prior studies (Hsu, 2004; Gulati et al., 2012). Detailed description of features for VC firms is presented in Table 4.

## D    AGENT CLUSTERS

In the PARTNERMAS workflow, for each experimental setting, the Planner Agent is prompted to generate a set of Specialized Agents. Specifically, it produces a list of profiles, one for each agent. Each profile specifies four elements: (i) agent name, (ii) role, (iii) abilities, and (iv) guides. Across all 14 experimental settings using the PARTNERMAS framework, a total of 9,786 profiles are generated. Since agent names can vary slightly across settings, grouping by name alone would yield an excessive number of fragmented clusters. To address this, we cluster agents using embeddings so that those within the same cluster share similar profiles, whether in roles or guides emphasizing comparable features of candidate VC firms. The clustering process proceeds as follows:

- We use an embedding model "all-MiniLM-L6-v2" to create one profile vector for each specialized agent.
- We then use the "k-means" clustering method to aggregate all agents. After experiments on the number of clusters, we find that $k = 8$ (i.e. eight clusters) balances the cluster size, similarity within clusters, and diversity between clusters, and achieve an acceptable Silhouette score at $0.290$. As a result, we obtain eight clusters for all 9,786 agent names among 14 test experimental setting.

## E    DEBATE MAS DESIGN

The Debate MAS enhances decision quality through structured multi-phase interaction among agents. Instead of relying on a single evaluation, the system encourages critique, reflection, and oversight, leading to more robust outcomes.

Table 4: Key features used in VC co-investor shortlisting and their descriptions.

| Feature | Type | Description |
|---|---|---|
| *Identifiers & labels* | | |
| companyid | ID | Unique identifier of the target company in the focal deal. |
| vcfirmid | ID | Unique identifier of the candidate VC firm. |
| leadvc | ID | Identifier of the lead VC for the focal deal. |
| real | Binary | Ground-truth label: 1 if candidate VC appears in the actual syndicate; 0 otherwise. |
| leadornot | Binary | Indicator for whether the VC is the lead investor. |
| yearquarter | Categorical | Year–quarter context of the focal deal (e.g., 2019Q3). |
| year | Numeric | Calendar year of the focal deal. |
| realsize | Numeric | Number of ground-truth co-investors in the focal syndicate. |
| *Target company attributes* | | |
| companyindustrymajorgroup | Categorical | Major industry grouping of the target company. |
| companynation | Categorical | Target company headquarters nation. |
| companystate | Categorical | Target company headquarters state (if applicable). |
| companycity | Categorical | Target company headquarters city. |
| companyzip | Categorical | Target company ZIP/postal code. |
| companylat | Numeric | Latitude of target company. |
| companylng | Numeric | Longitude of target company. |
| *Candidate VC firm attributes* | | |
| firmtype | Categorical | Type of investor. |
| firmnation | Categorical | VC firm nation. |
| firmstate | Categorical | VC firm state (if applicable). |
| firmcounty | Categorical | VC firm county (if applicable). |
| firmzipcode | Categorical | VC firm ZIP/postal code. |
| firmgeographypreference | Text | Stated geographic investment preferences. |
| firmindustrypreference | Text | Stated industry/sector preferences. |
| firminvestmentstagepreference | Text | Stated stage preferences (e.g., seed, early, growth). |
| *Candidate VC activity & outcomes (rolling/cumulative)* | | |
| vcfirm_dealcount_20qtr | Numeric | Deals by the VC in the past 20 quarters. |
| vcfirm_numcompinvest_20qtr | Numeric | Co-investments by the VC in the past 20 quarters. |
| vcfirmIPOcount_20qtr | Numeric | IPO exits associated with the VC in the past 20 quarters. |
| vcfirm_IPOcount_cum | Numeric | Cumulative IPO exits associated with the VC to date. |
| vcfirm_dealcount_cum | Numeric | Cumulative deals by the VC to date. |
| vcfirm_numcompinvest_cum | Numeric | Cumulative co-investments by the candidate VC to date. |
| *Network measures & pairwise history* | | |
| boncent | Numeric | Bonacich centrality of the VC in the co-investment network. |
| degree | Numeric | Degree centrality in the co-investment network. |
| pair_tie_strength | Numeric | Prior collaboration strength with the lead VC. |
| *Candidate VC geospatial* | | |
| uszip_vc | Categorical | U.S. ZIP code of the VC (normalized field, if applicable). |
| uslat_vc | Numeric | U.S. latitude of the VC office (normalized field). |
| uslng_vc | Numeric | U.S. longitude of the VC office (normalized field). |
| uscity_vc | Categorical | U.S. city of the VC office (normalized field). |
| uscounty_vc | Categorical | U.S. county of the VC office (normalized field). |

- **Evaluation Phase:** Each agent independently evaluates the candidate firms, producing initial scores and rationales.

- **Debate Phase:** Agents review their peers' reasoning (not the scores) and provide agreements, disagreements, and clarifying questions. This peer-review process emphasizes justification quality and helps surface biases or overlooked factors.

- **Reflection Phase:** After the debate, each agent revisits its own evaluation in light of peer feedback, reflecting critically and optionally adjusting its decisions.

In the end, a dedicated supervisor agent integrates the outcomes from all phases, resolves conflicts, and produces the final decision. The supervisor focuses on synthesizing insights across agents rather than simple score aggregation.

# F  PROMPT DESIGN

## F.1  BUSINESS DOMAIN GUIDANCE

We include the following business domain guidance to ensure that agents explicitly consider factors such as collaboration history, industry fit, strategic alignment, financial strength, and geographic proximity when making investment decisions:

```
Your decision should consider important dimensions like network and
    collaboration history (pair tie strength with the lead company and
    boncent), industry fit (firm industry preference), strategic
    alignment (firm investment stage preference), financial, and
    geography (distance, firm state). Your strategic guidance should
    explain which of these dimensions are most critical for this specific
     deal.
```

## F.2  PROMPT FOR SINGLE AGENT

```
You are '{self.name}', {self.role}, possessing {self.ability}.

# Your Profile:
{self.profile}

# Business Hint: (when enabled)
Your decision should consider important dimensions like network and
    collaboration history
(pair tie strength with the lead company and boncent), industry fit (firm
    industry preference),
strategic alignment (firm investment stage preference), financial, and
    geography (distance, firm state).
Your strategic guidance should explain which of these dimensions are most
    critical for this specific deal.

# Investment Target Company:
{target_profile}

# Lead Investor Profile:
{lead_profile}

# Candidate Co-Investors to Evaluate:
{candidates_list}

# Evaluate candidates across different dimensions based on the agent
    expertise and profile.

# Scoring Guidance (1-5 scale):
- 1: Poor match, significant concerns, not recommended at all
- 2: Below average, considerable issues, generally not favorable
- 3: Average neutrality, acceptable but with clear reservations
- 4: Good candidate, strong fit with minimal concerns
- 5: Excellent candidate, ideal investment partner and highly recommended

# Task:
Evaluate each candidate and select the top {top_k} co-investors for this
    investment opportunity.
```

### F.3 PROMPT FOR DEBATE MAS

**Stage 1: Initial Evaluation Prompt**

```
You are '{self.name}', {self.role}, possessing {self.ability}. Evaluate
    the following candidates for potential investment:

# Scoring Guidance (1-5 scale)
- 1: Poor match, significant concerns, not recommended at all.
- 2: Below average, considerable issues, generally not favorable.
- 3: Average neutrality, acceptable but with clear reservations.
- 4: Good candidate, strong fit with minimal concerns.
- 5: Excellent candidate, ideal investment partner and highly recommended
    .

# Investment Target:
{target_profile}

# Your Own Profile:
{self.profile}

# Candidates to Evaluate:
{candidates_data}

# Additional Context:
{context}

# Evaluation to be strictly JSON formatted:
{
  "evaluations": {
    "firm_id_1": {
      "integrity_score": int (1-5),
      "integrity_rationale": "... clear rationale why this score was given
          ...",
      "capability_score": int (1-5),
      "capability_rationale": "... clear rationale why this score was given
          ...",
      "fit_score": int (1-5),
      "fit_rationale": "... clear rationale why this score was given ..."
    },
    "firm_id_2": {
      // ... same structure for each firm
    }
  }
}
```

**Stage 2: Reflection Prompt**

```
You are '{self.name}', ({self.role}). After reviewing your evaluations:

Evaluations:
{evaluations}

Context:
{context}

Reflect critically on the evaluations. Provide clear thoughts in strictly
    JSON:
{
"reflection_summary": "... comprehensive reflection on possible biases,
    assumptions, or key insights ...",
"improvement_suggestions": ["clear suggestions on improvement", "... more
    thoughtful suggestion"],
"score_decisions": {
    "reasoning": "... your explicit self-reflect reasoning clearly ...",
```

```
    "stick_with_previous_score": true|false
}
}
```

**Stage 3: Debate Prompt**

```
You are '{self.name}' ({self.role}). You are reviewing evaluations
    written by your peer agents.

 IMPORTANT:
- You are debating with OTHER AGENTS about their evaluations, not with
    the firms being evaluated
- The numeric scores from your peers are intentionally hidden
- Focus ONLY on their **reasoning and justifications**  not the scores
- You are reviewing ONLY your peers' evaluations, not your own
- The supervisor's evaluations are not included in this debate

Available peer agents to debate with: {peers_list}

All Agents' Evaluations (scores hidden):
{stripped_evaluations}

Context:
{context}

Your task:
- Critically analyze the reasoning about the firms from other agents
- For each peer agent's evaluation, you can:
 * Agree with multiple points they made about a firm
 * Disagree with multiple points they made about a firm
 * Agree with some parts while disagreeing with others
- If something is unclear, ask specific questions directly to the
    relevant agent

Output as strictly formatted JSON:
{
  "agree": [
    {
      "agent_name": "name of the peer agent from available peer agents",
      "points": [
       "specific points you agree with about their evaluation",
      ]
    }
  ],
  "disagree": [
    {
      "agent_name": "name of the peer agent from available peer agents",
      "points": [
       "specific points you disagree with about their evaluation",
      ]
    }
  ],
  "questions": ["concise question directed clearly to agent_name about
      their evaluation"]
}
```

### F.4 PROMPT FOR PARTNERMAS (OURS)

**Planner Agent Prompt**

*Generic*

```
You are '{name}', a {role} with {ability}.
```

```
Your task is to design a multi-agent system for evaluating potential co-
    investor partnerships.
Based on the provided lead investor profile, target company profile and a
     sample of candidate co-investor profiles,
determine the optimal number of specialized agents, their specific roles,
     abilities, and profiles.

The goal is to create agents that can thoroughly evaluate candidates
    across different, relevant dimensions to find the best co-investors
    for the lead firm.
```

*Business-Domain-Guided*

```
You are '{name}', a {role} with {ability}.
Your task is to design a multi-agent system for evaluating potential co-
    investor partnerships.
Based on the provided profiles, determine the optimal specialized agents
    across different dimensions
(e.g., collaboration history, industry fit, strategic alignment,
    financial, geography, integrity) and formulate a high-level strategic
     guidance for the final decision-maker.

# Lead Investor Profile:
{lead_profile}

# Investment Target Profile:
{target_profile}

# Sample Candidate Co-investor Profiles (structure overview):
{sample_candidates}
(Note: This is just a sample of 2 candidates, infer general dimensions
    from the structure and target profile.)

# Your output MUST be a JSON object with TWO top-level keys: "
    strategic_guidance" and "agents".
# 1. "strategic_guidance": A concise paragraph outlining the most
    critical factors for selecting a co-investor for THIS SPECIFIC target
    . This is high-level advice for the supervisor.
# 2. "agents": A JSON array of agent configurations. Each agent must have
     a distinct profile that covers a key evaluation criterion inspired
    by the strategic guidance.
```

## Specialized Agent Prompt

```
You are '{name}', a {role} with {ability}.
Your specific focus is: {profile}.

# Investment Target:
{target_profile}

# Candidates to Evaluate:
{candidates_data}

Your task is to identify and rank the **top {dynamic_top_k}** most
    suitable candidate co-investor companies from the total of {
    total_candidates} candidates for the investment target.
Focus specifically on your area of expertise as defined in your profile
    using a clear logical flow:
# 1. **Select Focus:** State the key features you will focus on.
# 2. **Formulate Overall Strategy:** Explain your overall reasoning and
    methodology based on that focus.
# 3. **Make Decisions:** Rank the top candidates according to your focus
    and reasoning.
```

```
# Your Output MUST be a JSON object with THREE top-level keys in this
    specific order:
 - "evaluation_focus": A concise string identifying important features
     you are using for your analysis.
 - "overall_rationale": A general explanation of your ranking methodology
     , consistent with your stated focus.
 - "ranked_candidates": A list containing *exactly* the top {
     dynamic_top_k} candidates. Each rationale in this list must be a
     direct result of applying your focus and overall rationale.
```

**Supervisor Agent Prompt**

*Co-investor Selection by Importance*

```
You are '{name}', {role}, and you have the final say on co-investor
    selection.
Your goal is to produce a final, ranked shortlist of exactly **{top_k}**
    candidates.

# Strategic Guidance from Planner:
# This is the high-level strategy you must follow for this specific deal.
{planner_strategic_guidance}

# Your Decision-Making Process (Follow these steps precisely):
1. **Step 1: Identify Consensus Picks.**
    - Review all agent evaluations and identify candidates that are highly
        ranked by multiple agents.
    - Add the strongest consensus candidates to your shortlist first.
    - In your rationale, state how many consensus picks you found.

2. **Step 2: Fill Remaining Slots via Conflict Resolution.**
    - You now need to fill the remaining slots to reach the target of **{
        top_k}** candidates.
    - Examine candidates with mixed reviews (e.g., ranked high by one
        agent but low or not at all by another).
    - Use your Agent Importance Ranking as the decisive tie-breaker. The
        opinion of a more important agent carries significantly more
        weight.
    - Select the best of the remaining candidates based on this weighted
        analysis until your shortlist has exactly **{top_k}** members.
```

*Co-investor Selection by Weight - Weight Assign Prompt*

```
You are '{name}', {role}. Your goal is to make the best possible co-
    investor selection.
Before you review the candidate rankings from your specialized agents,
    you must first determine the numerical weight of each agent's
    perspective for this specific investment opportunity.

# Your Task:
Assign a numerical weight to each specialized agent based on how critical
    their focus is for this specific target. The weights must be a
    floating-point number (e.g., 0.35) and **the sum of all weights must
    equal 1.0**.
```

*Co-investor Selection by Weight - Selection Prompt*

```
You are '{name}', {role}, possessing {ability}. Your profile is: {profile
    }.
You are the General Partner and have the strongest voice in deciding who
    gets invited and joins the round.
Your task is to review the detailed evaluations from your specialized
    agents and, guided by the numerical weights you just assigned, make
    the final decision on the top {top_k} candidates for co-investment.
```

```
# Your Decision:
Based on all the information provided, and critically, *following the
    numerical weights you established*, select the best {top_k}
    candidates.
- For each candidate: Sum up the weights of all agents that recommended
    this candidate
- Final ranking: Order candidates by their total weighted scores from
    highest to lowest
```

*Co-investor Selection by Majority Vote*

```
 You are '{name}', {role}, possessing {ability}. Your profile is: {
     profile}.
You are the General Partner and have the strongest voice in deciding who
    gets invited and joins the round.
Your task is to review the detailed evaluations from your specialized
    agents and make the final decision on the top {top_k} candidates
    based on a **majority vote**.

# Your Decision:
Based on all the information provided, and select the best {top_k}
    candidates.
- Identify which candidates are most frequently recommended by the
    different agents.
- A candidate that appears on multiple agents' lists should be
    prioritized.
- Your final list should represent the collective decision from your team
     of agents.
```

