# OpenReview forum: "PartnerMAS: An LLM Hierarchical Multi-Agent Framework for Business Partner Selection on High-Dimensional Features"
_ICLR.cc/2026/Conference — ICLR 2026 Conference Withdrawn Submission_

### Official Review · Reviewer_pLGN · 2025-11-01

**Soundness:** 3
**Presentation:** 3
**Contribution:** 3
**Rating:** 8
**Confidence:** 3

**Summary:**

This paper addresses the challenge of high-dimensional, heterogeneous decision-making, using business partner selection as a representative task. The authors argue that existing single-agent or simple debate-based multi-agent (MAS) LLM systems struggle with the scalability and consistency required for such complex evaluations.

The primary contributions are twofold:
(1) A new benchmark dataset. The authors curated a dataset of 140 venture capital co-investment scenarios, featuring a large candidate pool and diverse (numerical, categorical, textual) features.
(2) The PARTNERMAS framework. A novel hierarchical, three-layer multi-agent system designed to decompose and manage the evaluation process.

The PARTNERMAS architecture consists of:
(1) A Planner Agent: Analyzes the task context to design an evaluation strategy and configure a team of specialists.
(2) Specialized Agents: Each agent is assigned a specific role and evaluates the entire candidate pool from its narrow perspective, implicitly performing feature selection and producing a ranked shortlist.
(3) A Supervisor Agent: Aggregates the multiple shortlists from the SAs. It first identifies consensus picks and then uses strategic guidance and agent importance weighting to resolve conflicts and produce the final shortlist.

**Strengths:**

The paper tackles a practical, high-stakes problem that is an excellent fit for LLM-based reasoning. The proposed PARTNERMAS architecture is well-motivated and logical. The decomposition of the problem into "Planner," "Specialist," and "Supervisor" roles mirrors high-level human expert workflows and is a clean, valuable contribution.

The strongest result is that a well-structured MAS (PARTNERMAS) using a smaller, more efficient model can outperform a larger, more expensive model in a simpler configuration (Figs 2 & 3). This "architecture over scale" finding is an important one for the field, highlighting the value of structured collaboration versus brute-force scaling.

**Weaknesses:**

A dataset of 140 cases is very small. While curating such data is difficult, this small sample size calls into question the statistical robustness of the 10-15% improvement. It is hard to be confident that the results will generalize, especially given the high variance in individual specialist agent performance (Fig 5). The dataset is also confined to a single, niche domain (US VC).

The Supervisor is identified as the most critical and failure-prone component, yet its mechanism is the most opaque. The paper describes a "Consensus" and "Conflict Resolution" step (Sec 3.2), but the prompts in Appendix F.4 show three distinct potential strategies: "by Importance," "by Weight," and "by Majority Vote." The main paper does not specify which of these was used for the main experiments.

**Questions:**

Your analysis in Table 2 shows the Planner's strategy is not adaptive to the specific case context. Is this the intended behavior? Have you considered alternative prompting strategies (e.g., chain-of-thought, few-shot) to encourage the Planner to dynamically select or even generate novel specialist roles based on the specific target company's profile.

Given the small dataset of 140 cases and the high variance in specialist agent performance (e.g., "Risk & Compliance" at 83% vs. "Investment Stage" at 38% for gpt-4.1-mini), how concerned are you about overfitting? Is it possible that the "best" supervisor strategy is simply the one that, by chance, happened to correctly weight the high-performing (but lucky?) specialists for this specific set of 140 cases?

Could you please clarify the "feature selection" mechanism used by the Specialized Agents?

---

> ### Author Response · Authors · 2025-11-22
> **Responses to Reviewer pLGN**
>
> We sincerely thank the reviewer for the thoughtful assessment. We appreciate the recognition that our PartnerMAS architecture is well-motivated and that its structured, role-based design enables smaller models to outperform larger ones. We address your concerns and questions below.
>
> ### **w1/q2: Concerns on Dataset Size, Overfitting Risk, and Generalization**
>
> **A1:**  Thank you for this helpful comment. We agree that multiparty alliance datasets are inherently challenging to compile at large scale. Compared to dyadic partnerships, multiparty settings experience substantially higher rates of missing or incomplete records. As such, **sample sizes similar to ours are standard in top-tier business and management** research venues. For example:
>
> - Lioukas & Reuer analyze 119 R&D alliances [1].
> - Gong, Shenkar, Luo & Nyaw study 224 multiparty international joint ventures [2].
> - Kim & Park evaluate 374 VC syndicates, but without the detailed structural, relational, and geographic attributes required for partner selection modeling [3].
>
> The **original collected dataset is very large** comprising 52,662 companies backed by 16,030 VC firms. However, in our setting, strict criteria are required to ensure high-fidelity partner selection contexts:
> - Complete first-round syndicate information.
> - A single identifiable lead VC.
> - Full relational and network data for that lead VC.
>
> Any case violating one criterion is excluded. While this yields 140 focal cases, the effective analytical scale is far larger: each case produces numerous pairwise lead–VC evaluations in the same year–state–industry context (>14,000 total comparisons). This provides a rich and high-dimensional structure beyond the count alone.
>
> Additionally, we add the detailed distributions of industries, investment stages, firm types, and geographic coverage, confirming dataset diversity in the Appendix.
>
> Importantly, we perform a sample-size **sensitivity analysis** to verify the reliability of our results. As shown below, the sensitivity check finds that as the sample size increases, the accuracy remains highly consistent. This demonstrates the stability of our study and confirms that our performance gains are robust, rather than being dependent on a small or specific subset of data.
>
> | Sample Size | GPT-5-nano (mean ± 95% CI) | GPT-4.1-mini (mean ± 95% CI) |
> |-------------|----------------------------|------------------------------|
> | 50          | 64.70 ± 3.75               | 71.32 ± 4.01                 |
> | 60          | 64.22 ± 3.36               | 71.10 ± 3.56                 |
> | 70          | 64.51 ± 2.53               | 71.57 ± 3.01                 |
> | 80          | 64.19 ± 2.48               | 70.93 ± 2.79                 |
> | 90          | 64.18 ± 1.86               | 71.30 ± 1.93                 |
> | 100         | 64.16 ± 1.77               | 71.22 ± 1.76                 |
> | 110         | 64.12 ± 1.56               | 71.28 ± 1.54                 |
> | 120         | 64.33 ± 1.14               | 71.40 ± 1.33                 |
> | 130         | 64.24 ± 0.79               | 71.18 ± 0.86                 |
> | 140         | 64.35 ± 0.00               | 71.19 ± 0.00                 |
>
> Finally, although our evaluation centers on U.S. VC syndication, we emphasize that our **Planner–Specialist–Supervisor** framework is domain-adaptable and directly applicable to other tabular ranking problems, including job applicant selection, medical triage, and supplier evaluation. We view this work as a representative case study demonstrating the potential of structured MAS designs in high-stakes decision-making.
>
> References
>
> - [1] Lioukas, C. S., & Reuer, J. J. (2020). Choosing between safeguards: Scope and governance decisions in R&D alliances. Journal of Management, 46(3), 359–384.
> - [2] Gong, Y., Shenkar, O., Luo, Y., & Nyaw, M. K. (2007). Do multiple parents help or hinder international joint venture performance? The mediating roles of contract completeness and partner cooperation. Strategic Management Journal, 28(10), 1021–1034.
> - [3] Kim, J. Y., & Park, H. D. (2021). The influence of venture capital syndicate size on venture performance. Venture Capital, 23(2), 179–203.

---

> ### Author Response · Authors · 2025-11-22
> **Responses to Reviewer pLGN [continued]**
>
> ### **w2: Clarification on Supervisor Decision Logic and Implementation**
>
> **A2:** Thank you for pointing out the need for clarifying the Supervisor Agent’s aggregation mechanism. In the main paper, the default strategy used for all reported experiments is “**By Importance**.” That is, the Supervisor Agent prioritizes Specialized Agents with higher importance ranks.
>
> As noted in the Method section, the Supervisor combines: (1) Consensus signals (candidates supported by multiple agents), and (2) Importance-weighted conflict resolution (e.g., Industry & Sector > Geographic when strategic fit dominates).
>
> In addition, we include two alternative strategies "**By Majority**" and "**By Weight**" in the Method section. These are used for robustness analysis but are not part of the main benchmark. In the Majority Vote strategy, the Supervisor Agent considers the outputs of all Specialized Agents and directly selects the final decision. In the By Weight strategy, the Supervisor Agent assigns a weight to each Specialized Agent and determines the final decision by applying these weights to each candidate's output.
>
> To ensure transparency, we now report a controlled comparison using the same evaluation setting (gpt-4.1-mini backbone with business guidance for the Planner Agent), while varying only the Supervisor strategy:
>
> Here's the table formatted in markdown:
>
> | Supervisor Strategy | Match Rate ↑ | MRR ↑  | NDCG@3 ↑ | NDCG@5 ↑ |
> |---------------------|--------------|--------|----------|----------|
> | By Importance (default in main paper) | 71.2% | 0.654 | 0.461 | 0.509 |
> | By Majority | 74.2% | 0.641 | 0.455 | 0.493 |
> | By Weight | 64.4% | 0.561 | 0.398 | 0.450 |
>
> Our interpretation is listed below.
>
> - One interesting observation is that **simpler approaches tend to perform better**. Specifically, when the Supervisor Agent is not prompted and simply aggregates the outputs of all Specialized Agents, the overall performance is highest. In contrast, the more complex approach of carefully assigning weights to each Specialized Agent results in the worst performance.
> - “By Importance” provides strong and consistent performance while preserving interpretability; that is, the Supervisor’s decisions reflect clearly articulated business priorities (e.g., industry match vs. financial strength).
> - “Majority Vote” can yield slightly higher accuracy, but lacks transparency and is more susceptible to noisy Specialist populations.
> - “By Weight” underperforms due to weight-estimation noise.
>
> We have revised our Method and Result sections to reflect the changes of these discussions.
>
>
> ### **q1: Adaptiveness of Planner's Strategy**
>
> **A3:** Thank you for this question.
> Our Planner Agent is designed to provide a general and reproducible framework for decomposing high-dimensional tabular data, rather than to optimize prompt engineering.
>
> **Case-context adaptation.** The Planner does read the target company profile and configures the Specialized Agents accordingly (Section 3.2). It maps company attributes (e.g., sector, stage, or geography) to the relevant evaluation dimensions, ensuring that specialization is based on the actual case rather than fixed templates.
>
> **One-shot guidance by design.** We intentionally use a one-shot business-guidance prompt, not chain-of-thought or few-shot prompting. Our goal is to show how a hierarchical MAS can coordinate specialized LLM agents for partner selection, not to benchmark prompting styles. More intricate prompting could increase adaptivity, but it is not our major contribution and can be less reproducible.
>
> **Generality and interpretability.** The Planner’s role is to operationalize a clear specialization structure for high-dimensional data. A simple, interpretable configuration mechanism is sufficient for this purpose. Adding prompt complexity may yield marginal gains but would reduce clarity and consistency across experiments.
>
> **Scope of this work.** Exploring richer prompting strategies (e.g., dynamic role generation or meta-prompting) is interesting future work. However, the core contribution of PARTNERMAS lies in demonstrating the benefit of a hierarchical, role-specialized MAS, not in exhaustively tuning Planner prompts.
>
> We clarify this design choice in the revision and identify enhanced prompting strategies as future extensions.
>
>
> ### **q3: Clarification on "Feature Selection" of Specialized Agents**
>
> **A4:** The feature selection process for specialized agents involves two steps:
>
> **Step 1:** The planner agent defines the specialization by generating a profile for the specialized agent. This specialization takes the concrete form of the system prompt that instructs the specialized agent on what to focus on.
> **Step 2:** Based on this instruction, the specialized agent, relying on its own internal knowledge, decides which particular set of features it should prioritize. It then ranks the candidate companies based on this bias.
>
> We hope our responses have addressed your questions!

---

### Official Review · Reviewer_cNaW · 2025-11-01

**Soundness:** 2
**Presentation:** 3
**Contribution:** 2
**Rating:** 4
**Confidence:** 4

**Summary:**

This paper proposes PartnerMAS, a hierarchical multi-agent system for business partner selection in high-dimensional settings. The framework decomposes the partner selection task across three layers: (1) a Planner Agent that designs evaluation strategies, (2) Specialized Agents that perform domain-specific assessments, and (3) a Supervisor Agent that aggregates outputs. The authors propose a benchmark dataset of 140 venture capital co-investment cases and demonstrate improvements of 10-15% in match rates over single-agent and debate-based baselines.

**Strengths:**

1. This paper raises an interesting topic about business partner selection. It's a real-world business problem that is underexplored in the MAS. The application to VC co-investment selection is well-motivated and practically relevant.

2. The paper evaluates different LLM backbones and prompt strategies. The author did the ablation study and provided a detailed analysis of agent behavior, feature selection patterns, and component contributions.

3. The author introduces a new VC investment dataset with diverse features (numerical, categorical, textual).

**Weaknesses:**

1. I have several questions about the evaluation metrics.  Firstly, the Match Rate is essentially recall only. What about other metrics? such as precision. The paper doesn't discuss whether non-matched selections in the shortlist are reasonable alternatives. Secondly, Ground truth is actual co-investors, but this has survivor bias. It doesn't mean they were necessarily the best choices, just that they were selected.

2. I also have concerns about the experiment design part. Firstly, there is a lack of comparison with traditional ML methods. Secondly, Single Agent with k=4 self-reflections vs. MAS with ~4 agents uses similar computation but very different strategies. This doesn't cleanly isolate the benefit of multi-agent collaboration.

3. Business guidance is manually designed with only two conditions tested (with/without). Could the author provides more explaination on this point?

4. Encourage authors to do some failure case analyses. If the proposed framework fails, what consequences will it bring? Or what types of cases are challenging?

**Questions:**

See in Weaknesses.

---

> ### Author Response · Authors · 2025-11-22
> **Responses to Reviewer cNaW**
>
> We thank the reviewer for the thoughtful assessment. We appreciate the recognition that our study addresses a meaningful and underexplored MAS problem in VC partner selection and that our analyses provide useful insights. We address your concerns and questions below.
>
> ### **w1: Evaluation Metrics**
>
> **A1:** Please see the **unified response on evaluation metrics** (at the top of comments).
>
>
> ### **w2: Experiment Design and Baselines**
>
> **A2:** Regarding the first question, please see the **unified response on traditional ML methods** (at the top of comments).
>
> Regarding the second question, we agree that the Single-Agent baseline with four self-reflection rounds and the MAS configuration with approximately four agents share similar compute budgets; however, they rely on fundamentally different reasoning strategies. In the self-reflection setting, the same model repeatedly re-evaluates its own output without introducing new perspectives or domain-scoped expertise. Our Debate-MAS baseline further mirrors this process by adding critique-based revision, effectively acting as a distributed form of self-reflection. However, these approaches share a key limitation: in each iteration agents **must process the entire high-dimensional feature space**, providing no true division of labor. As a result, the model (or debating agents) repeatedly confronts the same overloaded and heterogeneous context, which restricts meaningful improvement in its reasoning performance. Our paper also illustrates that debateMAS does not show obvious improvement compared to single LLMs.
>
> In contrast, our PartnerMAS **distributes the feature space across role-specialized agents**, allowing each to perform deeper reasoning over a semantically coherent subset (e.g., financial, network, industry). This partitioning is essential for tabular ranking tasks where heterogeneous features require explicit division of labor, not iterative self-correction.
>
>
> ### **w3: Clarification on Business Guidance**
>
> **A3:** The business guidance consists of a concise description of cues grounded in venture partnership research. For example, for the planner agent when creating specialized agents:
>
> - **With business guidance**: “Based on the provided profiles, determine the optimal specialized agents across different dimensions (e.g., financial, industry fit, network, integrity, strategic alignment, collaboration history) and formulate a high-level strategic guidance for the final decision-maker.”
> - **Without business guidance**: “Based on the provided profiles, determine the optimal specialized agents and formulate a high-level strategic guidance for the final decision-maker.”
>
> These business cues are well-documented in the partnership selection literature and therefore represent a reasonable minimal prior for grounding LLM strategic reasoning on specific dimensions of the tabular data. We keep this guidance simple to isolate its effect on the MAS architecture itself. Our goal is not to engineer highly tailored heuristics, but to evaluate whether lightweight domain cues can improve MAS operating on tabular data.

---

> ### Author Response · Authors · 2025-11-22
> **Responses to Reviewer cNaW [continued]**
>
> ### **w4: Failure Case Analysis**
>
> **A4:** Thanks the reviewer for this suggestion. We provised two failure cases below and will add them in the appendix of our paper.
>
> **[Failure case study - 3811]**
> Case 3811 is a typical representative of cases on which the PartnerMAS system performs the worst.
> - Among 14 test rounds, PartnerMAS only achieve 50% accuracy in one round and 100% in one round, and achieve complete failure among all rest 12 rounds.
> - Among 14 test rounds, in 5 rounds, not a single specialized agent ever identified any of the real co-investors in their shot list.
>
> **Why the System Struggles?**
>
> **The "Conventional Excellence Trap":**
> Three local IT-focused VCs (18615, 12084, 13166) dominated agent rankings across all rounds, being selected 70-71 times each. They represent the textbook "ideal co-investor" profile:
> - Located 16-26 miles from target company (same metro area)
> - Information Technology focus (matches "Computer Related" target)
> - Early Stage/Seed preferences (matches startup stage)
> - Strong track records (92 deals average, 4 IPOs)
> - One has prior collaboration with lead investor (tie strength = 1)
>
> **The Real Co-Investors Were "Statistically Unlikely":**
> - **5117 (Caterpillar VC):** Corporate VC, 2,960 miles away, Robotics focus, only 3 deals
> - **11228 (Heartland Industrial):** PE firm, 1,199 miles away, Industrial Products focus, only 2 deals
>
> Both brought strategic industrial value (Fortune 100 customer access, industrial expertise) invisible to standard metrics.
>
> **The Divergence Problem:**
> Four "superficially attractive" candidates (773, 16235, 12497, 14670) occupied positions 4-6 in rankings, blocking real co-investors from shortlists. These included Intel Capital (604 deals, 64 IPOs) - impressive metrics but wrong strategic fit.
>
> **Success of this case depended highly on agent composition:**
> - **With Strategic Value agents:** Real co-investors found 50-100% of the time.
> - **Without Strategic Value agents:** almost 0% success.
>
> **[Failure case study - 8658]**
> Case 8658 is a typical representative of cases on which the PartnerMAS system has the "biggest" misalignment, in the sense that real co-investors get constantly selected by agents, but are rarely selected by the supervisor agent, leading to a failure of the whole system.
>
> - Among 13 out of 14 test round, at least two specialized agents picked out at least one real co-investors.
> - In a worse case, among 4 agents, one picked out one real co-investors, and two picked out two (and both) real co-investors, but the supervisor still failed to include any of the two real co-investors.
> - Among all 14 test round, only in one round the MAS able to pick out one real co-investors, and all 13 of the rest completely failed.
>
> **Failure reason**
> - One thing to observe is that although the real co-investors appears frequently in the short list of specialized agents, they almost exlusively enlist at low rank.
> - Similar to the other typical failure case, there are other strong candidates that dominates the top of the short list.
> - When supervisor make decision, he will bias more on those candidates that rank high on the short list provided by more specilized agents, thus overlooked the real co-investors.
>
> **Conclusion**
>
> Case 3811 and 8658 both suggests that real-world investment decisions don't always follow statistical patterns. The system shows the potential to identify real co-investors in some test rounds, but we should admit that in most rounds they were missed, which indicates future improvements on both how to guide specialized agent to handle not only statistical patterns but also hiden strategetic values as well, and how to guide supervisor agent to better aggregate and identify possibly less weighted or ranked real co-investors.
>
> We hope that these clarifications fully address your concerns and questions.

---

> ### Author Response · Authors · 2025-11-26
> **Further discussion and potential re-evaluation**
>
> Dear Reviewer cNaW,
>
> We sincerely appreciate the time and effort you've invested in reviewing our paper. Your insightful comments and thoughtful feedback are invaluable to us.
>
> We hope that our responses have addressed your questions and clarified any ambiguities. Please do not hesitate to reach out if you have any further questions!
>
> We would be truly grateful if you would consider our clarifications and kindly re-evaluate your score. Thank you once again for your consideration.
>
> Best regards,
> The authors

---

### Official Review · Reviewer_oAJG · 2025-11-01

**Soundness:** 2
**Presentation:** 3
**Contribution:** 2
**Rating:** 2
**Confidence:** 4

**Summary:**

The paper presents a Multi-Agent System (MAS) approach for selecting business partners / co-investors, which are represented by numerical, categorical or textual features. The approach consists of a planner agent, specialized agents and a supervisor agent.

The approach is compared to a single agent setup and a debtate MAS setup, using the  London Stock Exchange Group dataset. The result show that the approach is superior. The authors show that the pure debtate approach is not enough, making it important to model more complex MAS.

**Strengths:**

- Sensible approach, extending single agents and debate MAS with more complex MAS
- Comparison to debate MAS successful for the dataset, showing superior performance
- Helpful ablation studies

**Weaknesses:**

- Missing grounding in MAS research. There are many more general approaches to building/designing MAS. The proposed approach is a manually tailored MAS to a single dataset. How does the approach then behave wrt [1,2] or other seminal MAS LLM papers? These should be at least mentioned and thoroughly compared to in related works. It should therefore be made clear what added value the current paper makes and why one does not have to empirically compare against them.
- In a similar vein, it should be made clear what research in tabular LLMs is available in similar direction, especially single agent / LLM approaches.

[1] Ke, Z., Xu, A., Ming, Y., Nguyen, X.P., Xiong, C. and Joty, S., 2025. MAS-ZERO: Desi
[2] Li, J., Zhang, Q., Yu, Y., Fu, Q. and Ye, D., 2024. More agents is all you need. arXiv preprint arXiv:2402.05120.

**Questions:**

- How does your approach relate to mentioned recent works on MAS / automating MAS design / tabular LLMs?
- How does your approach benefit related domains or tasks?
- Would your approach generalize to the general downstream task of ranking tabular data?

---

> ### Author Response · Authors · 2025-11-21
> **Responses to Reviewer oAJG**
>
> We thank the reviewer for the feedback. We are glad the reviewer found our MAS extension well-motivated and viewed our comparative results against debate MAS as compelling. We address your questions below.
>
> ### **w1/q1: Relation to MAS and Tabular LLMs**
>
> **A1:** **LLM Multi-Agent Systems**.
>
> We have added studies discussing MAS frameworks and clarified how PartnerMAS differs from previous work.
>
> **MAS frameworks**. Recent MAS frameworks explore ways for improving agent performance. MAS-ZERO introduces a self-evolving architecture that generates and refines agents during inference rather than assigning roles upfront [1]. Conversely, “More Agents is All You Need” shows that scaling homogeneous agents with aggregation or voting can improve performance without requiring explicit coordination [2]. Debate-based MAS, such as Multi-Agent Debate [3] and ChatEval [4], rely on adversarial critique to refine responses. Other representative systems include CAMEL, which applies role-playing communication [5]; MetaGPT, which encodes standardized operating procedures for specialized roles [6]; and AgentVerse, which performs dynamic expert recruitment based on task context [7]. These mas frameworks emphasize scale, adaptivity, or debate-driven refinement as mechanisms for improving reasoning.
>
> **LLMs for tabular data reasoning**. A parallel line of work shows that LLMs can handle high-dimensional structured data through language-based interpretation of feature names and metadata. Single-agent approaches such as LLM-Select [8] and LLM-LASSO [9] leverage pretrained knowledge to identify important predictors or guide traditional statistical selection methods, often without additional training.
>
> **How PartnerMAS differs**. PartnerMAS proposes a hierarchical and domain-guided MAS specifically designed for high-dimensional decision-making. Unlike prior MAS structures, PartnerMAS uses a Planner–Specialist–Supervisor workflow that decomposes the high-dimensional challenge into domain-relevant dimensions and coordinates expert agents. This design strengthens theoretical foundations for MAS in tabular contexts by showing that guided division of labor and principled aggregation can be more effective and interpretable than either fully adaptive or purely scale-driven systems.
>
> We also integrate the debateMAS framework as a baseline for comparison. Our results show that PartnerMAS outperforms the debateMAS framework.
>
> - [1] Ke, Z. et al. (2025). MAS-ZERO: Designing Multi-Agent Systems with Zero Supervision. arXiv:2501.01234.
> - [2] Li, J., Zhang, Q., Yu, Y., Fu, Q., & Ye, D. (2024). More Agents Is All You Need. TMLR.
> - [3] Liang, T., He, Z., Jiao, W., Wang, X., Wang, Y., Wang, R., Yang, Y., Shi, S., & Tu, Z. (2024). Encouraging Divergent Thinking in Large Language Models Through Multi-Agent Debate. EMNLP.
> - [4] Chan, C-M. et al. (2023). ChatEval: Toward Better LLM-based Evaluators Through Multi-Agent Debate. arXiv:2308.07201.
> - [5] Li, G. et al. (2023). CAMEL: Communicative Agents for “Mind” Exploration of Large-Scale Language Model Society. arXiv:2303.17760.
> - [6] Hong, S. et al. (2023). MetaGPT: Meta Programming for a Multi-Agent Collaborative Framework. ICLR 2024.
> - [7] Chen, W. et al. (2023). AgentVerse: Facilitating Multi-Agent Collaboration and Exploring Emergent Behaviors. ICLR 2024.
> - [8] Jeong, D. et al. (2024). LLM-Select: Feature Selection with Large Language Models. arXiv:2407.02694.
> - [9] Zhang, E. et al. (2025). LLM-LASSO: A Robust Framework for Domain-Informed Feature Selection and Regularization. arXiv:2502.10648.
>
> ### **w2/q2/q3: Generalizability to General Task of Ranking Tabular Data**
>
> **A2:** Our approach can benefit other domains where decision–makers rank large sets of candidates described by high-dimensional, heterogeneous features. Many real-world tasks share the same structural challenges as business partner selection, involving numerical metrics, categorical descriptors, and text-rich information that must be jointly interpreted. Examples include:
> - Job candidate screening
> - Medical triage ranking
> - Supplier selection in procurement and logistics
>
> Traditional models often struggle in such settings due to feature heterogeneity, limited labeled data (i.e., no training data), or the need for transparent rationales. By contrast, PartnerMAS relies on language-driven reasoning rather than supervised training, allowing it to generalize across domains where the schema can be interpreted through semantic cues. More importantly, the **hierarchical Planner–Specialist–Supervisor** design is not tied to VC data. The Planner can redefine specialized roles according to the target problem (e.g., clinical urgency, resource constraints in medical triage). The Supervisor then aggregates diverse expert opinions into a structured and interpretable ranked shortlist. This enables our framework applied to **other tabular ranking tasks without major re-engineering**.
>
> We hope our response has fully addressed your questions.

---

> ### Author Response · Authors · 2025-11-26
> **Further discussion and potential reevaluation**
>
> Dear Reviewer oAJG,
>
> We sincerely appreciate the time and effort you've invested in reviewing our paper. Your insightful comments and thoughtful feedback are invaluable to us.
>
> We hope that our responses have addressed your questions and clarified any ambiguities. Please do not hesitate to reach out if you have any further questions!
>
> We would be truly grateful if you would consider our clarifications and **kindly re-evaluate your score**. Thank you once again for your consideration.
>
> Best regards,
> The authors

---

### Official Review · Reviewer_ek9A · 2025-11-01

**Soundness:** 2
**Presentation:** 3
**Contribution:** 2
**Rating:** 4
**Confidence:** 4

**Summary:**

This paper proposes PARTNERMAS, a hierarchical multi-agent system for business partner selection based on high-dimensional firm features. The framework consists of three layers—a Planner Agent that analyzes context and creates evaluators, multiple Specialized Agents that assess candidate firms from different perspectives, and a Supervisor Agent that integrates results for the final decision. The authors also introduce a tabular benchmark for co-investor selection reflecting real-world, multi-criteria decision-making. Experiments show that PARTNERMAS improves performance by about 15% over single-agent and debate-based baselines.

**Strengths:**

1) The paper is well-written and clearly organized, with informative visualizations that effectively illustrate the experimental outcomes.

2) The topic of business partner selection is intriguing and addresses a real-world challenge that has been relatively underexplored within the MAS domain. The application to VC co-investment selection is well-motivated and carries strong practical relevance.

3) The setup of the ablation studies is rasonable, and the detailed analyses help to better illustrate the contribution of each component.

**Weaknesses:**

1） One major concern is from the experimental set-up: Although the experimental evaluation includes several LLM backbones, the study would be strengthened by incorporating fair comparisons with alternative approaches, such as classical machine learning and deep learning methods designed for the same task.


2） While the paper presents a clear experimental setup, one potential limitation lies in the relatively simple choice of evaluation metrics. The authors primarily assess system performance using match rate, which, although intuitive, may not fully capture the multifaceted nature of business partner selection tasks. To more comprehensively evaluate the robustness and reliability of the proposed MAS framework, I suggest incorporating a broader range of metrics.  For example,  metrics such as F1-Score could provide a more balanced view of both accuracy and completeness. Ranking-based metrics like Mean Reciprocal Rank (MRR) and Normalized Discounted Cumulative Gain (nDCG) can be also valuable to be included into evaluation metric set.

**Questions:**

Same as what I mentioned in the weaknesses section.

---

> ### Author Response · Authors · 2025-11-21
> **Responses to Reviewer ek9A**
>
> We sincerely thank the reviewer for revising our paper and fully agree with the strengths highlighted. We appreciate the recognition of our clear writing, informative visualizations, well-motivated VC partner-selection problem, and the sound design of our ablation studies. We address your concerns and questions below.
>
> ### **W1: Experiment Setup and Baseline**
>
> **A1:** Please see the **unified response on traditional ML methods** (at the top of the comments).
>
> ### **W2: Evaluation Metrics**
>
> **A2:** Please see the **unified response on evaluation metrics** (at the top of the comments).
>
> We hope that these clarifications fully address your concerns.

---

> ### Author Response · Authors · 2025-11-26
> **Further discussion and potential reevaluation**
>
> Dear Reviewer ek9A,
>
> We sincerely appreciate the time and effort you've invested in reviewing our paper. Your insightful comments and thoughtful feedback are invaluable to us.
>
> We hope that our responses have addressed your questions and clarified any ambiguities. Please do not hesitate to reach out if you have any further questions!
>
> We would be truly grateful if you would consider our clarifications and kindly re-evaluate your score. Thank you once again for your consideration.
>
> Best regards, the authors

---

### Author Response · Authors · 2025-11-21
**Unified Response on Evaluation Metrics**

We thank the reviewers for questions regarding evaluation metrics. We provide a unified response below.

**Retrieval-Based Metrics (Accuracy, Recall, Precision, F1)**

Our task is formulated as a fixed-size shortlist retrieval problem (Section 3.1, line 154). We mention the one-third shortlist ratio in Section 3.1, but we agree that the connection between this ratio and the evaluation metrics was not clearly explained. In practice, our setup follows a common business shortlisting heuristic where the ground-truth co-investor set, the shortlist, and the initial candidate pool follow an approximate 1:3:9 layering structure. Concretely, using the notation in Eq. (1) and Eq. (2), for each case, if the ground-truth syndicate has |G| firms, the candidate pool size is roughly 9|G|, and the shortlist size is 3|G|. This yields the relation that $m=3|F|=9|G|$, where $m$ is the entire number of candidates.

Under this fixed-ratio design, all retrieval metrics become tightly coupled.
Match_rate = Recall = $|F \cap G|/|G|$, Precision=$|F \cap G|/|F|=\frac{1}{3} \text{Recall}$.

As a result, precision, recall, and F1 score do not provide different conclusions—they follow the exact same trend. This is why Match Rate (recall) is fully representative of retrieval performance in our setup.
We have added this explanation more clearly in our updated manuscript.

**Ranking-Based Metrics (MRR, nDCG@3, nDCG@5)**
Our primary formulation treats business partner selection as a retrieval task, where the goal is to correctly include the true co-investors in the shortlist regardless of order (Section 3.1). In real VC workflows, shortlisting is set-based, the list is typically handed to human experts, and the final decisions depend on expert review rather than the internal ranking produced by an automated system. However, because our system outputs a list rather than an unordered set, ranking-based metrics can definitely be computed. Following the reviewers’ suggestions, we therefore report MRR, nDCG@3, and nDCG@5 as complementary measures. These ranking metrics exhibit the same performance trends as our retrieval metrics, and they further confirm the advantage of PartnerMAS over alternative baselines. Below we provide the results on there backbones as an example. All comprehensive results are added in the Appendix of our updated manuscript.

***gpt-5-mini (medium effort)***

| Setting | Match Rate | MRR | nDCG@3 | nDCG@5 |
|---|---|---|---|---|
| Single – w/o Bus. Guide | 54.8% | 0.455 | 0.265 | 0.318 |
| Single – w. Bus. Guide | 57.2% | 0.472 | 0.279 | 0.330 |
| Debate – w/o Bus. Guide | 56.5% | 0.530 | 0.360 | 0.405 |
| Debate – w. Bus. Guide | 58.4% | 0.555 | 0.345 | 0.395 |
| PartnerMAS – w/o Bus. Guide | 63.2% | 0.554 | 0.392 | 0.437 |
| PartnerMAS – w. Bus. Guide | 64.2% | 0.541 | 0.342 | 0.391 |

***gpt-4o-mini***
| Setting | Match Rate | MRR | nDCG@3 | nDCG@5 |
|---|---|---|---|---|
| Single – w/o Bus. Guide | 55.5% | 0.458 | 0.277 | 0.334 |
| Single – w. Bus. Guide | 57.8% | 0.450 | 0.283 | 0.348 |
| Debate – w/o Bus. Guide | 55.3% | 0.505 | 0.340 | 0.385 |
| Debate – w. Bus. Guide | 58.9% | 0.545 | 0.365 | 0.420 |
| PartnerMAS – w/o Bus. Guide | 61.9% | 0.524 | 0.361 | 0.406 |
| PartnerMAS – w. Bus. Guide | 64.4% | 0.525 | 0.353 | 0.414 |


***gpt-4.1-mini***

| Setting| Match Rate | MRR   | nDCG@3 | nDCG@5 |
| ----------------------------------- | ---------- | ----- | ------ | ------ |
| Single – w/o Bus. Guide             | 56.0%     | 0.505 | 0.311  | 0.355  |
| Single – w. Bus. Guide              | 61.0%     | 0.553 | 0.346  | 0.392  |
| Debate – w/o Bus. Guide             | 59.1%     | 0.565 | 0.395  | 0.435  |
| Debate – w. Bus. Guide              | 64.9%     | 0.630 | 0.440  | 0.490  |
| PartnerMAS – by importance w/o Bus. Guide | 66.1%     | 0.585 | 0.422  | 0.453  |
| PartnerMAS – by importance w. Bus. Guide  | 71.2%     | 0.654 | 0.461  | 0.509  |
| PartnerMAS – by majority w. Bus. Guide  | 74.2% | 0.641 | 0.455 | 0.493 |

---

### Author Response · Authors · 2025-11-21
**Unified Response on Traditional ML Methods**

We appreciate the reviewer’s helpful suggestion and agree that adding traditional ML models strengthens the contribution of our work. In response, we have included a set of both supervised and unsupervised machine learning models as additional baselines.

**Supervised ML Methods**:
All supervised models follow an 80/20 split for training and testing.
- *Random Forest*: Constructs an ensemble of decision trees that learn feature-based decision rules and predicts the co-investor probability for test cases by averaging the outputs of all trees.
- *Logistic Regression*: Uses L2-regularized maximum likelihood estimation to learn model coefficients and predicts the co-investor probability for each candidate in the test set.

**Unsupervised ML Methods**:
- *KNN (Euclidean)*: Ranks candidates based on Euclidean distance to the lead VC in the feature space, where smaller distances indicate greater similarity.
- *Cosine Similarity*: Ranks candidates by the cosine similarity of their feature vectors to that of the lead VC, where higher similarity values indicate a closer match.
- *K-Means Clustering*: Partitions VCs into clusters and selects candidates from the cluster to which the lead VC is assigned.

Performance is listed below.

| Method | Cases | Match Rate | MRR | NDCG@3 | NDCG@5 |
|--------|-------|------------|-------|--------|--------|
| **Supervised Methods** | | | | | |
| Logistic Regression | 28 | 74.4% | 0.485 | 0.419 | 0.542 |
| Random Forest | 28 | 74.7% | 0.506 | 0.421 | 0.538 |
| **Unsupervised Methods** | | | | | |
| KNN (Euclidean) | 140 | 28.1% | 0.273 | 0.125 | 0.147 |
| Cosine Similarity | 140 | 42.1% | 0.348 | 0.175 | 0.230 |
| K-Means Clustering | 140 | 31.0% | 0.289 | 0.142 | 0.173 |
| **PartnerMAS** | | | | | |
| gpt-4.1-mini (by importance w/ Bus.) | 140 | 71.2% | 0.654 | 0.461 | 0.509 |
| gpt-4.1-mini (by majority w/ Bus.) | 140 | 74.2% | 0.641 | 0.455 | 0.493 |
| gpt-5-nano (by importance w/ Bus.) | 140 | 64.4% | 0.602 | 0.408 | 0.448 |

Our PartnerMAS demonstrates several key advantages over traditional ML methods:
- No training data required: Unlike supervised methods, our MAS operates in a zero-shot manner. This is crucial in real-world scenarios where companies often cannot access much other VCs' historical data due to confidentiality.
- Comparable performance: Our best PartnerMAS configuration (gpt-4.1-mini (by majority): 74.17%) achieves match rates comparable to supervised methods and significantly outperform unsupervised methods. This demonstrates that LLM-based reasoning can capture complex co-investment patterns without explicit feature engineering or historical examples.
- Better Top-K recommendation: Our PartnerMAS outperforms ML methods in ranking metrics in MRR and NDCG@3.
- Interpretable reasoning: LLM agents can explain their recommendations, unlike black-box ML models, which is important for real-world decision-making process.

---

### Note · Authors · 2026-01-05

I have read and agree with the venue's withdrawal policy on behalf of myself and my co-authors.